# Intensification of Hate Speech, Based on the Conversation Generated on TikTok during the Escalation of the War in the Middle East in 2023

José-Luis González-Esteban [1,*], Carmen Maria Lopez-Rico [1], Loraine Morales-Pino [2] and Federico Sabater-Quinto [1]

1 Department of Social and Human Sciences, Miguel Hernández University, 03202 Elche, Spain; carmen.lopezr@umh.es (C.M.L.-R.); fsabater@umh.es (F.S.-Q.)
2 Department of Social Sciences and Humanities, Ibero American University, Tijuana 22500, Mexico; lmorales@umh.es
* Correspondence: jose.gonzalez@umh.es

**Abstract:** The present research has been carried out concurrently with the conversation that took place on the social network TikTok during the most recent escalation of the war between Hamas and Israel in the Middle East (Gaza-Palestine) during the month of October 2023. The main objective of this article is to analyze of how young audiences are informed about complex problems, the quality of that information, and the consequences of the intensification of uncontrolled hate speech. Regarding the methodology, data were extracted from TikTok using the open-source tool tiktok-hashtag-analysis—hosted on GitHub—which facilitated the analysis of hashtags within the posts collected from this social network, starting with an initial sample of 17,654 comments. The article draws and reaches conclusions related to the fact that young audiences indeed are interested in the escalation of the conflict in the Middle East, as it is evident that the conversation—which is polarized—on TikTok about this issue has escalated considerably. Similarly, analysis of the extracted and filtered sample shows that the variable "hate speech" intensified on the platform during the analyzed conversation.

**Keywords:** TikTok; social media; polarization; hate speech; Islamophobia; Israel; Palestine





## 1. Introduction

This research is based on a general approach agreed upon by the academy: the transformations of intermediation. What previously could only be achieved through intermediaries can now be carried out independently thanks to new technologies. Consolidated behaviors related to the use and consumption of information in young audiences, increasingly distant from traditional media and very assimilated to social networks, have changed. In this media ecosystem, networks themselves evolve and gain audiences on the basis of technological advances and cultural and geographic issues that push one or the other to the forefront of the infotainment business (Reig et al. 2019). Digitalization has meant a paradigm shift in the task of intermediation for both the media and politicians, and "citizens have become indiscriminate when it comes to the consumption of information" ["ciudadanos se han vuelto promiscuos en el consumo de información"] (Sánchez-Cuenca 2022). The crisis of democracy, the rise of populism, the increase in violence, and hate speech can only be understood and analyzed in this context of the disintermediation of the public sphere. This line of thought has been defended by social scientists such as journalist and Nobel Peace Prize winner Maria Ressa (2022). She asserts that the system of checks and balances is breaking down because the technological companies that now control this media ecosystem—linked to the incipient political ecosystem—have given audiences a false sense of authority or sense of freedom and have allowed polarization, radicalization,

and political extremism to spread on social networks. Ryszard Kapuscinski (2007) defined "otherness" in an ideal situation as that public space in which the "other" ceased to be synonymous with the "unknown" and the "hostile"—with mortal danger and the incarnation of evil—in which each individual found within themselves a part, however miniscule, of that "other", or at least believed that they had and lived according to that knowledge. On the contrary, as pointed out by Innerarity (2022), the current situation strengthens the theory of political populism in a space that is disintermediated—paradoxically stitched together news-wise—in which technopopopulism (Bickerton and Invernizzi-Accetti 2021) offers technical solutions to problems and seeks, in theory, not to become ideological but rather to serve the people. Associated with information usage and consumption, populism may be related to the consumption of the scandalous, the shocking, the unprecedented, or the controversial, and inextricably linked to strategies of disinformation and manipulation. We start, therefore, in this research from this general approach on the transformations associated with digitalization that generate the conditions for a generalized questioning of the intermediation that affects the public sphere: politics and media.

In this general context, the particular study of the behavior of young audiences is of special interest (USinRED 2023) because it focuses on the study, analysis, and understanding of how young people stay informed and the use they make of social networks for this purpose, distinguishing between voluntary and involuntary consumption, and trying to establish the information stage for these young audiences in this new paradigm. In this sense, and going to the concrete, in this research we are mainly interested in knowing the use, consumption, and conversation of these audiences about complex political contexts, such as the current episode of violence and war in the Middle East. We intend to evaluate the quality of the media diet (media literacy) of these users based on the type of consumption observed and their level of training. One of the founding fathers of Web 2.0, who coined the term "virtual reality" ["realidad virtual"], argues emphatically that social networks make politics impossible, have alienated young people from democracy, and have been "enormously deadly weapons" ["armas enormemente mortíferas"] in recent international conflicts (Lanier 2018). Identifying which sources can be considered benchmarks according to the type of news being accessed (politics) and the fluctuations in interest based on the current affairs followed in the news is an objective in scientific research that also seeks to determine young people's level of knowledge about the media ecosystem: the identification of the disseminator and their possible agendas, the news source, the use of sources, the sociopolitical context, and their ability to make good use of the news content available on their social networks. The ecosystems have changed, and technology, the internet, and networks have transformed productive processes, but the structural filters concerning access to the sphere of public opinion, by means of bureaucratic deformations of the structures of public communication or by means of control and/or manipulation of the information flows, have been maintained, or indeed increased (Habermas 1981). In this same vein, recent studies such as those by Palau and López-García (2022) point out that, in a context of uncertainty and crisis, of fragmented public spheres, and devoid of alternatives that ensure a possible dialog, it is necessary to open a social debate in which the quality of news remains central.

And in this contextual framework described on information uses and consumption by young audiences, we intend to focus on the hate speeches (Azman and Zamri 2022) that proliferate and increase in public conversation on social networks. The United Nations considers hate speech to be any form of communication by spoken word, in writing, or through behavior that is an attack or uses derogatory or discriminatory language in connection to a person or group on the basis of who they are, or in other words, on the basis of their religion, ethnicity, nationality, race, color, descent, gender, or other factor of their identity (United Nation 2019). The main objective of those who create, promote, or disseminate hate speech is the dehumanization of the other. The "other" is one whom anyone can denounce, despise, hurt, or kill with impunity (Emcke 2017). The author investigates the rise throughout Europe (extensible to the rest of the planet) of parties or

movements that practice aggressive populism, fostering a climate of fanaticism on which this study aims to focus, understanding that hatred can only be combated through careful and slow observation, constant clarification, and scientific questioning. The Spanish penal code also regulates this issue through Organic Law 10/1995, which came into effect in 1996 and establishes that those who publicly encourage, promote, or incite—directly or indirectly—hatred, hostility, discrimination, or violence against a group, against a part of a group, or against a specific person because of their membership in that group for racist, anti-Semitic, or other reasons related to ideology, religion or beliefs, family situation, belonging to an ethnic group, race or nation, birth origin, sex, sexual orientation or identity, gender, illness, or disability would be committing a hate crime.

Taking into account the general object of study that interests us in this research: conversations with conditions of hate in the framework of a young social network during the upsurge of a political conflict between two very disparate actors (Israel and Palestine), the variable of ideology (Córdoba Hernández 2011) as a distorted and distorting vision of reality that allows the well-off group to strengthen that structural superiority and maintain the victims' subordinate identity is of undoubted interest. In the case of hatred, Cortina (2017) argues that there is a structural hierarchy in which the aggressor occupies the superior place while the aggressed against occupies the inferior one. This variable allows her to develop the term "aporophobia" as a hate crime based on the contempt and rejection of those who are worse off on the economic and social scale and the propensity to take a position in daily life and in conflicts that favors the better off—those from whom some benefit can be obtained—and to leave the poorest unprotected. This specific variable, not a minor one, is of undeniable interest for a conflict like the one at hand, which can be the subject of further research. Studies, such as Immigrationism (Red Acoge 2022), with a focus on (poor) migrants (Arcila-Calderón et al. 2022) and criticism of media praxis, warn about dehumanization and the need to incorporate approaches based on human rights. In short, the other, the different, is re-victimized when he is also poor, as is evident in research that has studied similar phenomena, such as the case of the gypsy community (Magano and D'Oliveira 2023), who are exposed to hate speech in conversations (comments) on social networks owing to their condition of poverty and social exclusion.

*TikTok, as a Case Study of Conversations in Young Audiences*

The specific case study focuses on the social network TikTok, the newest on the scene, which has excited audiences, revolutionized languages and storytelling, and magnified existing issues related to political conversation on established networks such as X (formerly Twitter) and Facebook and Instagram (Meta). Very young audiences have joined this social network over the last five years, and protecting minors from hate speech is of particular concern. The social platform presents itself to its audience as the destination "to capture and present the world's creativity, knowledge, and precious life moments, directly from the mobile phone. TikTok enables everyone to be a creator and encourages users to share their passion and expression through their videos". To achieve their goals, the designers worked on an algorithm with a nature that makes it easier to go viral. The scientific literature emerging from the research on this incipient communication phenomenon indicates that these positive, almost idealistic approaches and objectives do not always live up to these ideals. Research, such as that of Weimnn and Masri (2020), focusing their analysis on the propagation of racist, homophobic, ultranationalist, ultrareligious, and anti-Semitic messages from accounts linked to extremist organizations based on white supremacism and neofascism, and research such as ours confirm the toxicity of the conversation around a topic as internationally relevant as the conflict between Israel and Hamas. Originally, when the app launched in China in 2016 and during its meteoric expansion to the rest of the world, users were limited to sharing short videos that could be edited by using a variety of simple and absolutely innocuous creative tools associated with music, dance, and/or humor. With this formula and the slow response of its competitors in the United States, its growth has been explosive, as it established itself in a very short time and

consolidated its position as the social network with the most followers in the world in the age group between 16 and 24 years (Omnicore 2023), something that also occurred among Spanish-speaking audiences, where it is mostly used by an audience under 17 years of age. In 2023, 48% of its users were between 12 and 24 years old, and 57% were under 34 (Statista 2023a). Along these lines, there is unsettling information (Ortutay 2023) regarding open judicial investigations, such as the recent lawsuit that 42 of the 50 states in the United States brought against Meta, accusing the business of damaging young people's mental health and worsening the youth mental health crisis by designing/redesigning the algorithms of their platforms Instagram and Facebook to make them more addictive for minors, something that has been demonstrated in studies conducted around the world, such as the experiment conducted by USinRED (2023) with students from the last two years of middle school, from high school, and from the first years of university in Malaga, Elche (Alicante), and Madrid. In addition to the undeniable danger of addiction, some studies point out that children and/or adolescents are not prepared to maintain virtual relationships/conversations because they do not have the capacity to comprehend or assess concepts of privacy, and they are not aware of the risks that come with sharing photos or videos or, of course, making sense of a given context. Along these lines, Contreras (2022) concludes that the platforms have developed algorithms that create a polarization in society in the political, religious, or ideological spheres—without rules, without clear regulation, or without limits—that, among the youngest, are a breeding ground for lack of respect and intolerance toward the other (the different). Martínez Valerio (2023), in contrast, takes a positive view of some media outlets' work monitoring the content of their TikTok accounts, allowing for a nontoxic conversation, whereas Civil et al. (2023) highlight this application as a space for reshaping the dominant hate speech associated with Islam and terrorism.

This breakneck development and the generation of toxic content led TikTok to develop its terms of service to control and mitigate this toxicity as much as possible. In said contract with its users, the app states that users may not bully or harass another, nor promote sexually explicit material, violence, or discrimination on the basis of race, sex, nationality, disability, sexual orientation, or age; nor post any material that is defamatory of any person, obscene, offensive, pornographic, hateful, or inflammatory; any material that constitutes, encourages, or provides instructions for a criminal offense, dangerous activities, or self-harm; any material that is deliberately designed to provoke or antagonize people, especially trolling or bullying, or is intended to harass, frighten, distress, embarrass, or annoy; and any material that contains a threat of any kind, including discrimination on the basis of race, religion, gender, disability, or sexuality. These terms were later supplemented by what TikTok calls community standards that arise in the face of a proliferation of user behavior (content) that incites hate (Castaño-Pulgarín et al. 2021), of which the company said "this is not compatible with TikTok's creative and inclusive environment" ["no son compatibles con el entorno creativo e inclusivo de TikTok"] (TikTok 2021). At this point, they decided to develop a specific space to counteract hatred, which, as academic research on the issue shows, is not entirely effective. In this space, the application defines hate speech as behavior that incites hatred, attacks, threatens, dehumanizes, or denigrates a person or group of people because of their characteristics, including race, ethnicity, origin, nationality, religion, caste, sexual orientation, sex, gender, gender identity, disease, disability, and immigration status. Shortly before the emergence of TikTok, the Council of Europe (Comisión Europea Contra el racismo y la Intolerancia (ECRI) and Consejo de Europa (2017)) warned that editorial controls were lacking and that, although codes of conduct or ethics existed, monitoring was not always conducted systematically. Focusing on the use and consumption of social networks, the Council and the European Parliament themselves updated the legislation (European Union 2022), emphasizing that the digital transformation and social networks' explosive use and development carry with them ever more risks and challenges for the individual targets of the related services, companies, and society as a whole. Conclusions such as that of Nachawati (2020) on the consumption of social networks in relation to the conflict in the Middle East warn of the domination

of polarization based on hate speech, whereby the voices defending human rights are weakened and overshadowed, and sectarianism, the dehumanization of the other, and the narrative of a world divided into two extremes gain public space in the online conversation.

Based on this theoretical framework, we pose the following research questions, which we intend to answer with the methodological analysis carried out:

RQ1: Is there interest among young TikTok audiences in information about complex problems and international conflicts such as the Middle East between Israel and Hamas? RQ2: Has polarization and hate speech been generated on TikTok in the conversation in Spanish generated by the conflict in the Middle East between Israel and Hamas?

## 2. Method

The main objective of this study is to analyze the hate speech in the conversation that took place using TikTok comments in Spanish, during the recent escalation of war between Israel and Hamas, after the latter's attack on Israeli territory on 7th October. The sample runs from 2nd October until 20th October, three days after Israel's bombing of the Gazan hospital.

For TikTok data mining, we used the open-source tool tiktok-hashtag-analysis[1]—hosted on GitHub—which facilitated the analysis of hashtags within the collected TikTok posts (Borkhetaria et al. 2023). In an initial exploration, we focused on videos posted between 2nd October and 20th October 2023, prioritizing those considered most relevant, such as those appearing on the search page for each hashtag mentioned. The 1034 videos processed included at least one of the examined hashtags of interest: Hamas, Israel, Palestine [Palestina], Jews [judíos], Zionists [sionistas], Muslims [musulmanes], unaccompanied foreign minor [mena], Moor [moro], Jovenlandia (a fictional land from which immigrants are supposed to come, based on the tendency of the Spanish press to refer euphemistically to immigrants as "youths" or "jovenes"), ham [jamón], Arabs [árabes], Gaza, and Islam.

The information collected from the platform was structured in JSON format. We specifically selected the fields ID (unique identifier of the video), likes (number of likes), collect_Count (number of times saved), comment_Count (number of comments), playCount (number of plays), sharedCount (number of shares), share UrL (URL for sharing), hashtags (hashtags used), description (description of the video), author_id (unique identifier of the creator), author nickname (username of the creator), and creation_time (time created) to facilitate subsequent analysis. We cleaned the data, eliminating duplicate video entries, and built a table in .CSV format. This table was the initial database for the current study. Subsequently, we formulated a heuristic to assign a relevance value to each video, based on a weighting of different metrics associated with video interaction and visibility. These metrics are likes, playCount, comment_Count, and sharedCount, which represent the number of likes, plays, comments, and shares, respectively. The relevance formula we propose is:

$$\text{Relevance} = w1 \times \text{likes} + w2 \times \text{playCount} + w3 \times \text{comment\_Count} + w4 \times \text{sharedCount}$$

where w1, w2, w3, and w4 are weights reflecting the relative importance of each metric. For this initial heuristic, the following weights were assumed: w1 = 1 (for likes), w2 = 0.5 (for playCount), w3 = 2 (for comment_Count), and w4 = 3 (for sharedCount), based on the assumption that a comment indicates a higher-level interaction than a like and that sharing the video represents the most valuable interaction.

Comparing the metrics commonly used to assess the relevance of videos, we found that the number of views is a standard metric for determining video reach. We also consider the play rate to be important, viz., the percentage of unique visitors who played the video, and engagement metrics such as likes and comments. These metrics and the weighting proposed in the heuristic are consistent with common practices for assessing the relevance of a video, although the exact assignment of weights may vary depending on the context and objectives of the analysis. Additionally, we conducted a detailed analysis on the

frequency of occurrence of the hashtags associated with the videos from the study period. To do this, we built a table in which the first column represents the days, and the first row represents the hashtags. Each cell reflects the number of times a given hashtag appeared in the videos on a specific day. We included all hashtags present in the videos, not just those specific to the study, which provides a broader picture of the context of hashtags on the platform. The data were visualized using graphs illustrating the evolution of hashtag frequency over time. This representation allowed us to observe how the hashtags were positioned and related to each other, offering valuable insights on the interaction dynamics and trends on the platform during the period analyzed.

Owing to the inherent limitations of the "web scraping" technique, it was not possible for us to specifically control for the nationality of the content creators in the initial sample. Therefore, we chose to select a random sample of 27 videos, produced by young Spaniards, with the aim of analyzing user comments and assessing the prevalence of hate speech. For this purpose, we identified specific Spanish slang terms used pejoratively toward migrants of Muslim origin, such as "Jovenlandia", "they don't eat ham" ["no comen jamón"], and "Moors" ["moros"]. We extracted comments from each video using the exportcomments.com platform, which provides details such as the date and time each comment was made, the number of likes received, and the comment's textual content. After processing the 27 randomly selected videos, we generated a comprehensive database that included a total of 17,654 comments. These were coded and subjected to a comprehensive analysis using ATLAS.ti 23 software, applying co-occurrence analysis techniques and carrying out a sentiment analysis.

To perform a qualitative analysis of the comments on the discussions of these videos, we made sure that they had at least 60 likes, since we knew that such statements had considerable support from viewers. In total, 220 comments were analyzed, taking into account: the meaning to which it refers; if there is hate speech (Islamophobia, anti-Semitism, or none); if there is a position with the conflict (pro-Palestinian, pro-Israel, or neutral); and yes, there is a religious connotation (which may also have warlike or pacifist overtones) or pacifist-secular overtones in the statement.

For this, we have used discourse analysis, which is a qualitative methodology that allows us to study language as a social practice. It focuses on the analysis of the meanings that are constructed in discourses, taking into account the contexts in which they are produced (Van Dijk 1997).

In this article, discourse analysis is used to study hate speech in comments on TikTok videos with the goal of identifying the types of hate speech that occur on this platform as well as the contexts in which they occur.

Discourse analysis is an appropriate methodology for this study for the following reasons:

(1) It allows language to be analyzed in its social and cultural context. Hate speech is a social phenomenon that occurs in a specific context. Discourse analysis allows us to identify the social and cultural factors that contribute to the production of this type of discourse;

(2) It is flexible and can adapt to different types of speeches. Hate speech can be expressed in different ways. Speech analysis allows us to identify the different types of hate speech that occur on TikTok;

(3) It is a methodology that has been widely used to study hate speech. Discourse analysis has been used in numerous studies on hate speech in different contexts (Van Dijk 1993; Khan et al. 2019).

As stated by several authors, such as Cleary et al. (2014) or Baker (2006), the samples in discourse analysis do not necessarily have to be as broad as in quantitative methods. In fact, large samples can be a drawback in discourse analysis, as indicated by Bondarouk, T., and Ruël, H.J.M.:

> "Sample size is not usually a main issue in discourse analysis as the interest is in the variety of ways the language is used (Potter and Wetherell 1987). Large

variations in linguistic patterning can emerge from a small number of people. So a larger sample size may just make the analytic task unmanageable rather than adding to the analytic outcomes". (Bondarouk and Ruël 2004, p. 8)

Potter and Wetherell also influence this idea: "small samples or a few interviews are generally quite adequate for investigating an interesting and practically important-range of phenomena. For discourse analysts the success of a study is not in the least dependent on sample size. It is not the case that a larger sample necessarily indicates a more painstaking or worthwhile piece of research. Indeed, more interventions can often simply add to the labor involved without adding anything to the analysis" (Potter and Wetherell 1987, p. 161). So, this sample gives us a first approximation of what was being talked about (on Tiktok in Spanish) in the first moments of the conflict breaking out.

## 3. Results

In the first stage, we decided to explore the frequency of content production about the Hamas–Israel conflict using the scraping technique. To this end, we explored the main hashtags posted between 2nd October and 20th October 2023, to evaluate whether there was an increase in posts. According to the results observed, starting on October 7 (Hamas's attack on Israeli territory), there was a significant increase in the number of mentions of Israel, Hamas, Gaza, Palestine (in both English and Spanish, "Palestina"), FYP (For You page), and war in TikTok content, which indicated a very significant increase in discussion of the conflict on the aforementioned social network, reaching its peak on 11th October 2023 (Figure 1).

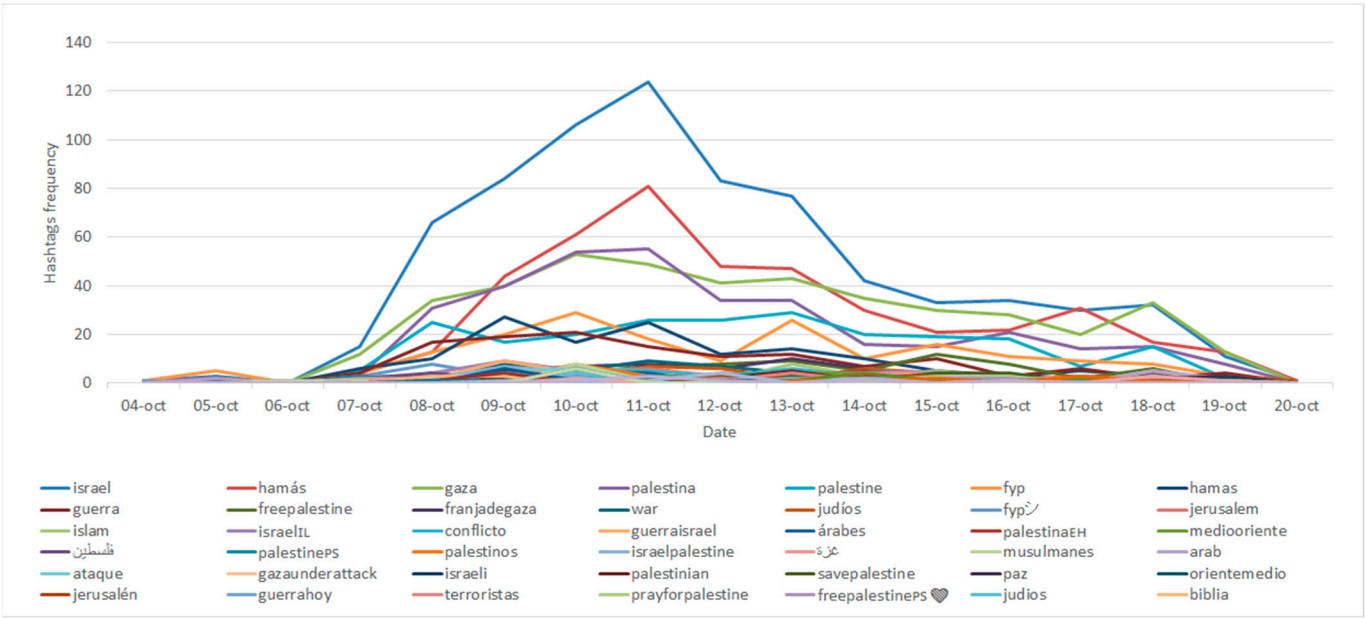

**Figure 1.** Daily frequency of hashtags referring to the Israel–Palestine conflict, between 4th October and 20th October 2023. Source: The authors' own creation based on the scraping technique.

As of 11th October, the frequency of posts about the Hamas–Israel conflict began to decrease considerably, and by the end of the analysis period—20th October—values similar to those at the beginning of the period under analysis were reached. This reduction in the frequency of posts suggests that interest in or coverage of the conflict online declined toward the end of the period in question (Figure 1).

After randomly selecting the 27 videos and processing the 17,654 comments, we proceeded to analyze word frequency to gain insight into the language used in conversations about the Israel–Hamas conflict, capturing both the emotional and factual aspects of the ongoing conflict. The data obtained reveals a number of significant patterns and observa-

tions that reflect the complexity and diversity of conversations around conflict and offer valuable insight into how people express themselves on this particular platform.

First, the presence of words related to the conflict, such as "war" ["guerra"], "attack" ["atacar"], "aggression" ["agresión"], "military" ["ejército"], and "dead" ["muertos"], highlighted the centrality of the topic of armed conflict in the conversations and the concern for the consequences and the actors involved. The appearance of words expressing intense opinions and emotions, such as "hate" ["odio"], "bad guys" ["malos"], "lie" ["mentira"], "worse" ["peor"], and "fear" ["miedo"], highlighted the emotional charge that this topic evokes in the audience. The use of emojis, such as 😒, 🙏, 🙄, and 😭, showed that conversations on TikTok also involve elements of emotion and opinion. Emojis can add an additional layer of expression and allow users to convey their feelings more directly (Figure 2).

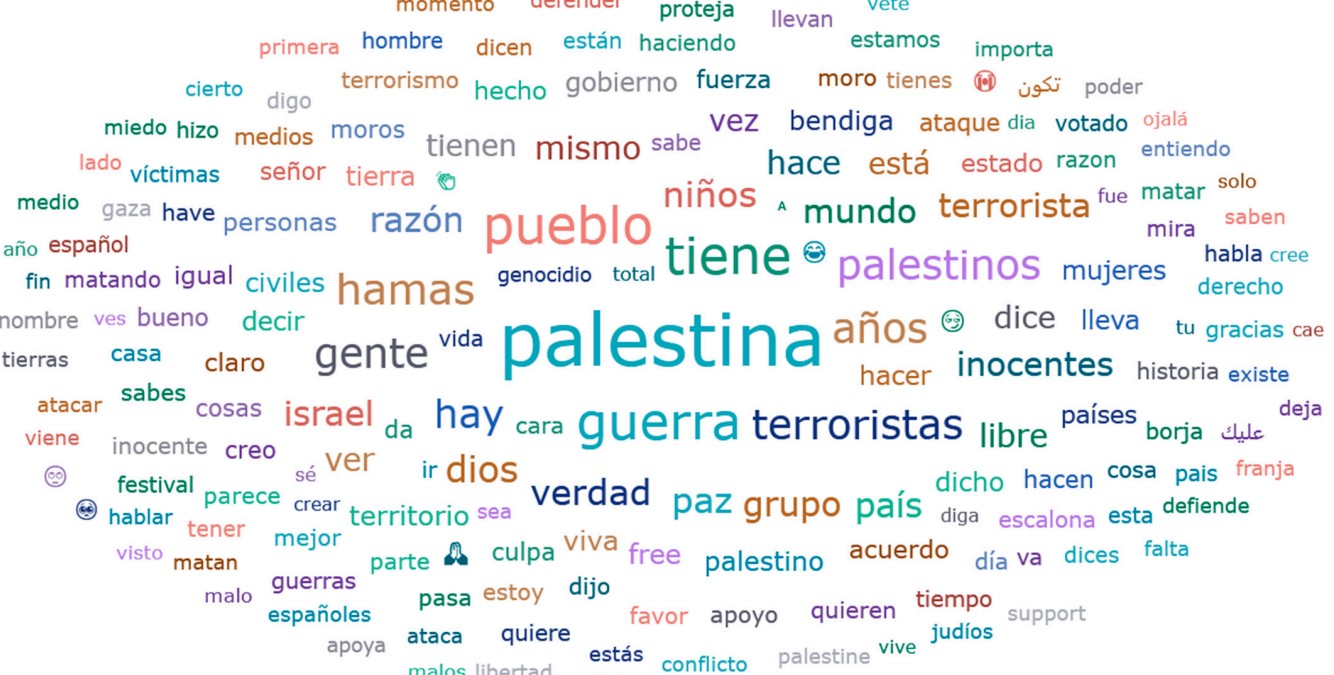

**Figure 2.** Analysis of the most common terms in comments on randomly selected TikTok videos. Source: The authors' own creation based on ATLAS.ti analysis.

The inclusion of words related to religion and culture, such as "Muslims" ["musulmanes"], "bible" ["biblia"], "Jews" ["judíos"], and "religion" ["religión"], suggested that the discussions also addressed issues of religious identity and belief, which is a key aspect of this conflict. The mention of words such as "right" ["derecho"], "defend" ["defender"], and "defend itself" ["defenderse"] pointed to the importance of legal and ethical aspects in the ongoing debate, including the discussion about self-defense (Figure 2).

Referencing words indicating support or criticism reflected the variety of perspectives and opinions at play. This underscores the diversity of voices and the controversy surrounding the conflict. The appearance of terms expressing uncertainty or confusion suggested that some people were seeking greater understanding or additional information on the subject, hence the importance of the media in disseminating and understanding events. Finally, the presence of words in different languages, such as Spanish, English, and Arabic, highlighted the linguistic diversity of the conversations and the participation of people from different cultural backgrounds, underlining the globality and scope of the conflict.

To process all the content generated from the 17,654 comments collected in the 27 sample videos, we decided to use the automatic coding tool based on artificial intelligence (AI) offered by the ATLAS.ti software (Lopezosa et al. 2023). After purging redundant

or repeated codes, we obtained a total of 2527 codes. We then used indicators such as co-occurrence and density to identify discursive associations between the codes or terms most used by participants in TikTok conversations, with the aim of identifying the most influential discursive elements in public opinion (Morales Pino 2023).

Initial analysis revealed that opinion code was related to sarcasm—with an emphasis on mockery and irony—as well as religious beliefs and support for one side of the conflict or the other. However, a recurring factor was the presence of verbal aggression among users and toward content creators. In this context, the opinion code also encompassed elements of comparison with, disagreement with, and explicit support for Palestine (Figure 3).

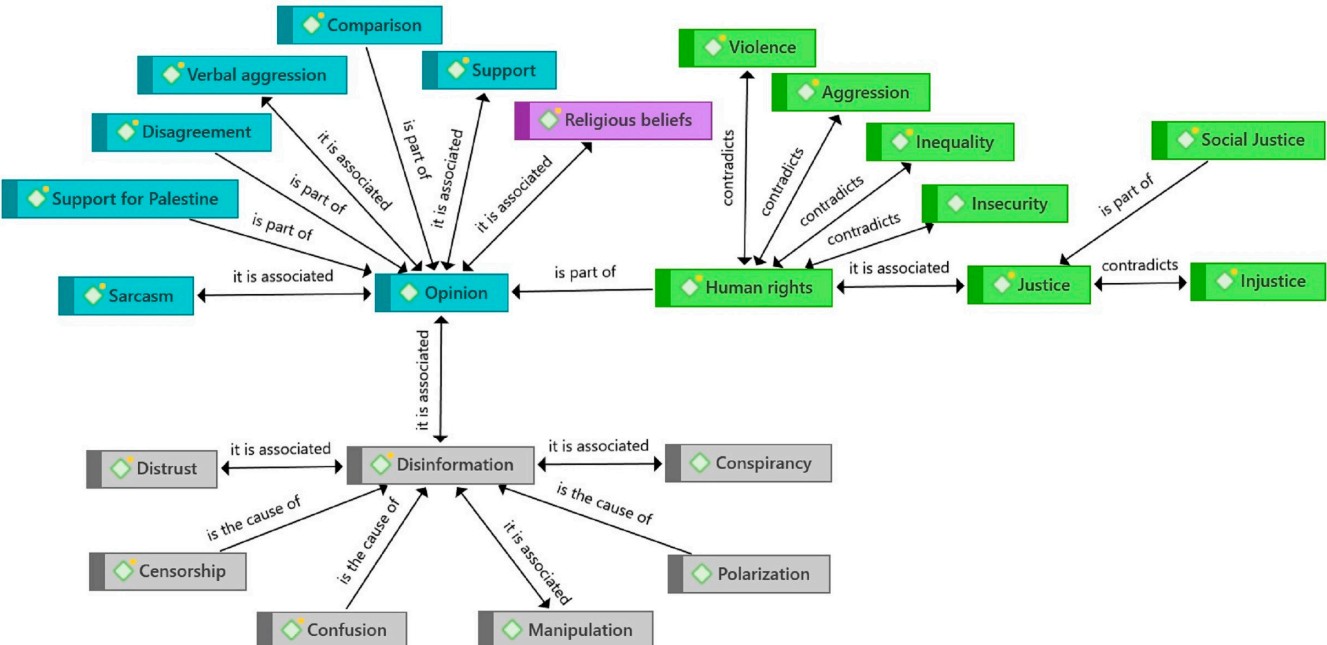

**Figure 3.** Influences on the opinions of TikTok users who commented on the 27 randomly selected videos. Source: The authors' own creation.

In addition, opinion code was also closely linked to disinformation, based on elements such as conspiracy, censorship, confusion, polarization, manipulation, and public distrust. Moreover, this was manifested in concerns related to human rights, which contrasted with the presence of verbal aggression and other types of violence. Furthermore, it was associated with messages that reflected inequality, insecurity, and violence, but also with the notion of justice, especially in the context of social justice (Figure 3).

In this context, complexities and contradictions in public opinion are evident, where seemingly opposing factors coexist, creating a multifaceted picture that reflects the diversity of perspectives and concerns in the TikTok conversations related to the conflict between Israel and Hamas.

As for the qualitative analysis, we observed (Figure 4) how the comments analyzed in the sample had as subjects different actors in the Palestinian–Israeli conflict, among which Palestine (28.7%), Israel (15.7%), and Hamas (12%) stand out. We also found comments alluding to both Hamas and Israel, the United States, and Israel or Palestine and Hamas, although always in a lower percentage. However, it is noteworthy that 20.8% of the comments alluded to Islam, and, as shown by Figure 5, in these cases, generally with a xenophobic tone.

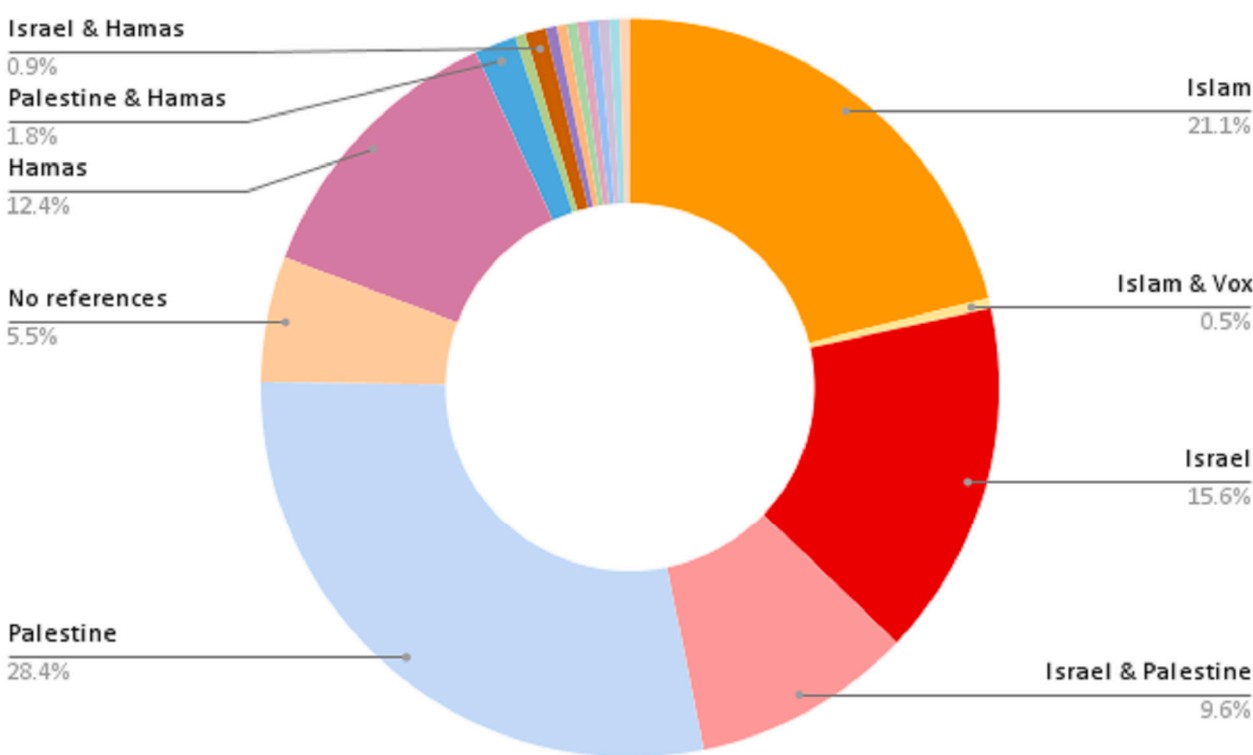

**Figure 4.** Who the comments talked about. Source: The authors' own creation.

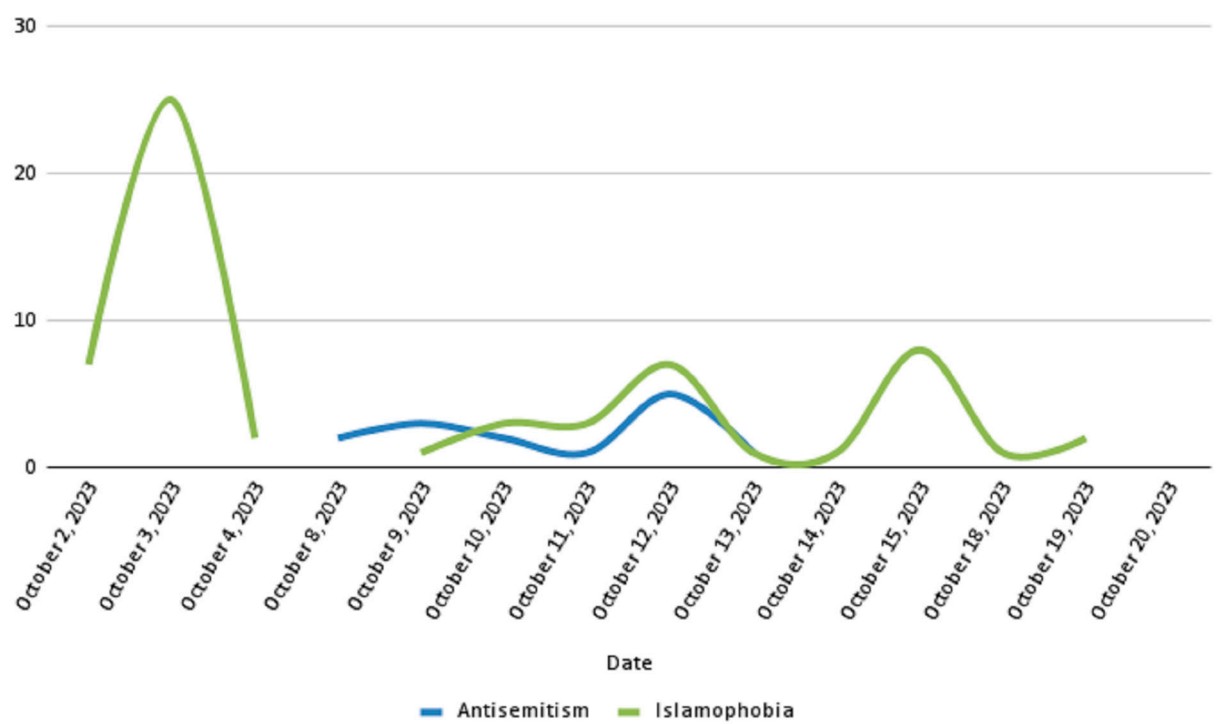

**Figure 5.** Timeline of the presence of xenophobia in the comments. Source: The authors' own creation.

One-third of the comments analyzed (34.1%) had xenophobic speech and were significantly Islamophobic (27.5%) and, to a lesser extent, anti-Semitic (6.9%). As shown in Figure 5, we found a peak in Islamophobia before the Hamas attack on 7th October; from this moment on, xenophobic speech was directed at both Muslims and Israelis, in which comments were polarized between the actors in the conflict, and refocused against Islam

on 15th October, which was paradoxically when the Palestinians in Gaza were suffering the most destruction.

Despite the fact that Spanish public opinion has had a pro-Palestinian tendency, as we already commented above, after 9/11, the world has seen Muslims as the enemy of Western culture, and in recent years, we have found hate speech against both Islam (religion) and Arabs (race), as sometimes the public is not clear about the difference between the two.

Many Islamophobic messages in the sample were not related to the conflict in Gaza itself but were posted from accounts with videos about immigration in Spain, to which followers responded and related the situation directly or indirectly to the conflict in Gaza:

-   "Let more immigrants from Jovenlandia come 🖌" ["Qué vengan más jóvenes de jovenlandia 🖌"].
-   "Why don't they go to Morocco to do the same thing?... Because they would cut off their ◯◯ (testicles), while in Spain they give them benefits 🙄" ["¿Por qué no se irán a Marruecos a hacer lo mismo?...por qué les cortan los ◯◯ (testículos) y en España les dan paguitas 🙄".].
-   "Crack down. They should pay for the destruction with hard labor, jail, and deportation without the possibility of ever setting foot on Spanish soil again" ["Mano dura, que paguen los destrozos con trabajos forzados, cárcel y expulsión sin posibilidad de volver a pisar suelo español".].
-   "Spain can build a mega prison and start acting like Bukele (El Salvador) and feed all these Moors pork, and if they don't want it, then let them die" ["España ya puede hacer una mega cárcel y empezar hacer como Bukele (El Salvador) y a todos éstos moros les daba de comer cerdo y si no lo quieren pues a morirse toca"].
-   "How much more do we have to put up with? Will we have to take measures to put an end to these people, I think? 🙄🙄🙄🙄" ["Qué más hay que aguantar?, habrá que tomar medidas para acabar con esta gente, digo yo?🙄🙄🙄🙄"].

Another example of this type of content was reflected in a video from the account sayyad_officiell, in which a Moroccan boy encouraged the youtuber Borja Escalona, known for spreading hate messages around different Spanish groups and regions, to explain to him "why you don't like Moors" ["por qué no le gustan los moros"], to which Escalona replied that he did not like "those who do not know how to behave in a country that has opened its doors to them" ["los que no se saben comportar en un país que les ha abierto las puertas"], and the boy told him that he was insulting them and that he should go and say what he thought to the face of a group of compatriots who were right there, and Escalona, saying that he had no problem telling them, walked away. In this video, we found the comment in the sample that had the most likes (10,663): "One of the few things I agree with Borja Escalona about" ["De las pocas cosas que estoy de acuerdo con Borja Escalona"], which makes it clear that a large number of TikTok users agree with this statement, which falls into the patterns of hate speech and incitement to violence.

We also found numerous direct allusions to Gaza or Palestine that had Islamophobic overtones, such as:

-   "Go over to Palestine, to enjoy the wonderful Islamic caliphate that awaits you there" ["Vete a Palestina, para disfrutar del maravilloso califato islamico que te espera allí"] (644 likes).
-   "Go over to the Gaza Strip, then tell me how women's rights are" ["Vete a la franja de Gaza, y después me cuentas qué tal los derechos de las mujeres"] (108 likes).

Some comments referenced hoaxes that have circulated and which many citizens—and even US President Joe Biden, who assumed that Hamas had beheaded babies during its attack—have fallen for. Even though the journalist rectified and clarified the confusion, some citizens continue to use this as a reason behind their position on the conflict: "Defending yourself is slitting babies' throats and murdering civilians? You all are sick in the

head" ["Defenderse es degollar bebés y asesinar civiles? Estáis enfermos de la cabeza"] (880 likes).

Regarding anti-Semitic messages, we found some, such as:

-   "What a pity that Palestine has no military power because Israel is the real terrorist for the genocide it is carrying out against Palestine" ["Una pena que Palestina no tenga potencia militar porque Israel es el auténtico terrorista del genocidio que está llevando a cabo contra Palestina"] (1315 likes).
-   "They have been murdering Palestinians for years, and nobody says anything. And now they feel offended. It disgusts me. FREE PALESTINE" ["Llevan años asesinando a palestinos y nadie dice nada. Y ahora ellos se sienten ofendidos. Asco es lo que me da. FREE PALESTINA"] (1556 likes).

Beyond the analysis of messages with xenophobic content that can be framed as hate speech and which represented almost one-third of the total analyzed, as we highlighted above, in Spain the majority has historically had sympathy toward Palestine, and this can be seen in the total sample of comments analyzed. We saw a predisposition for supporting Palestine in the conflict (41.7%), whereas pro-Israel comments were in the minority (18.3%), and only 11.9% remained neutral in their opinions on the matter. It should be noted that 28% of the comments did not refer to the conflict and therefore did not express a clear position on it (Figure 6).

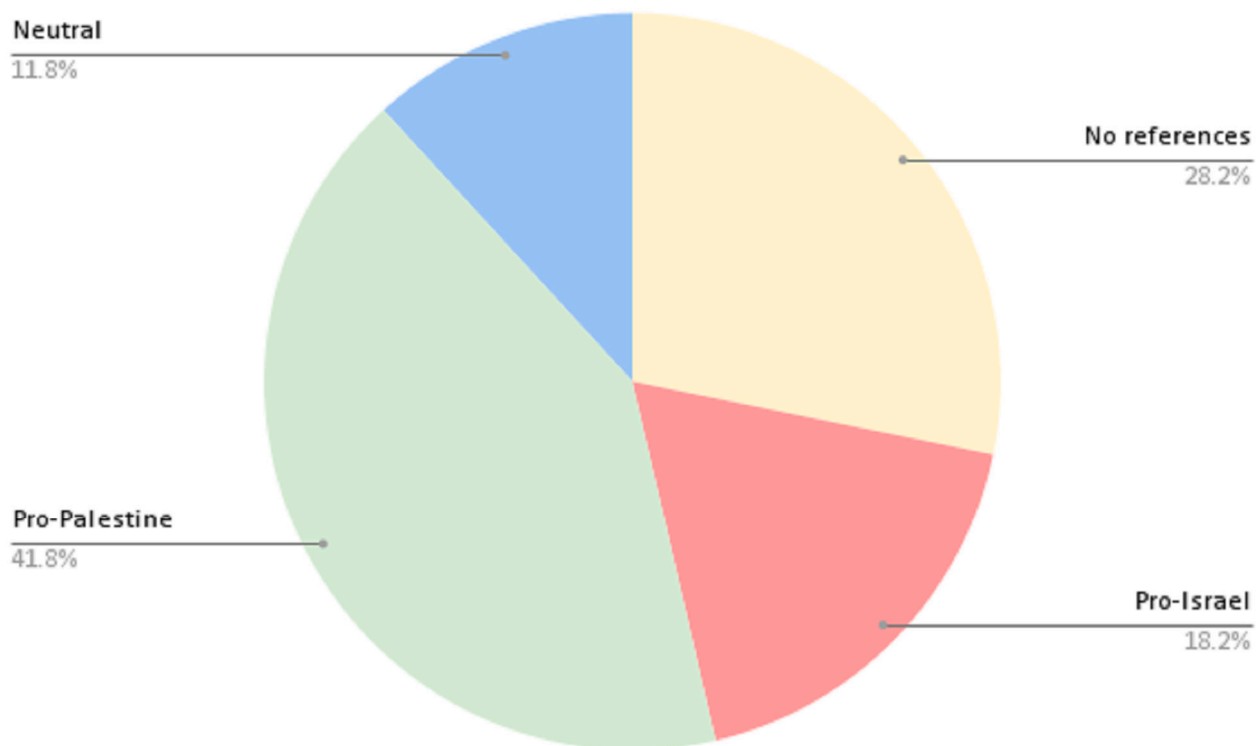

**Figure 6.** Positioning of comments on the Israel–Palestine conflict. Source: The authors' own creation.

Primarily, users indicated their support for Palestine using the slogan "Free Palestine" ["Palestina libre"], although there were other more specific or clarifying comments such as: "It's just that it's not Palestine. It's Hamas 🤦‍♀️, and they need to get better informed" ["Es que no es Palestina. Es Hamas 🤦‍♀️ y tienen que informarse mejor"]. Regarding those who took a pro-Israeli stance, they claimed that: "The world is with Israel" ["El mundo está con Israel"] or "TO TOP IT OFF, THEY WANT US TO SUPPORT PALESTINIA???? THEY ARE THE WORST, BLOODTHIRSTY AND SOULLESS" ["ENCIMA QUIEREN QUE APOYEMOS A PALESTINA???? SON LO PEOR, SANGUINARIOS SIN ALMA"]. Those who had a neutral opinion pointed out that: "In war there are no winners; everyone loses

😖😖" ["En la guerra no hay ganadores, todos pierden 😖😖"]. Some of the statements (11.9%), moreover, had a tone that was pacifist/secular (2.1%) "🙁 Peace for Israel, Palestine and the whole world. . ." ["🙁 Paz para Israel, Palestina y el mundo entero. . ."] or religious (4.9%) "God bless Palestine" ["Dios bendiga Palestina"], including religious with a warmongering or vengeful angle (2.1%) "Do not touch God's people" ["El pueblo de Dios no se toca"] or religious seeking peace and the end of the conflict (2.8%) "protect Israel and Palestine. Thank you Beloved Father. All are your children. Loving and merciful Father" ["protege a Israel y a Palestina, gracias Padre Amado, todos son tus hijos. Padre amoroso y misericordioso"] (Figure 7).

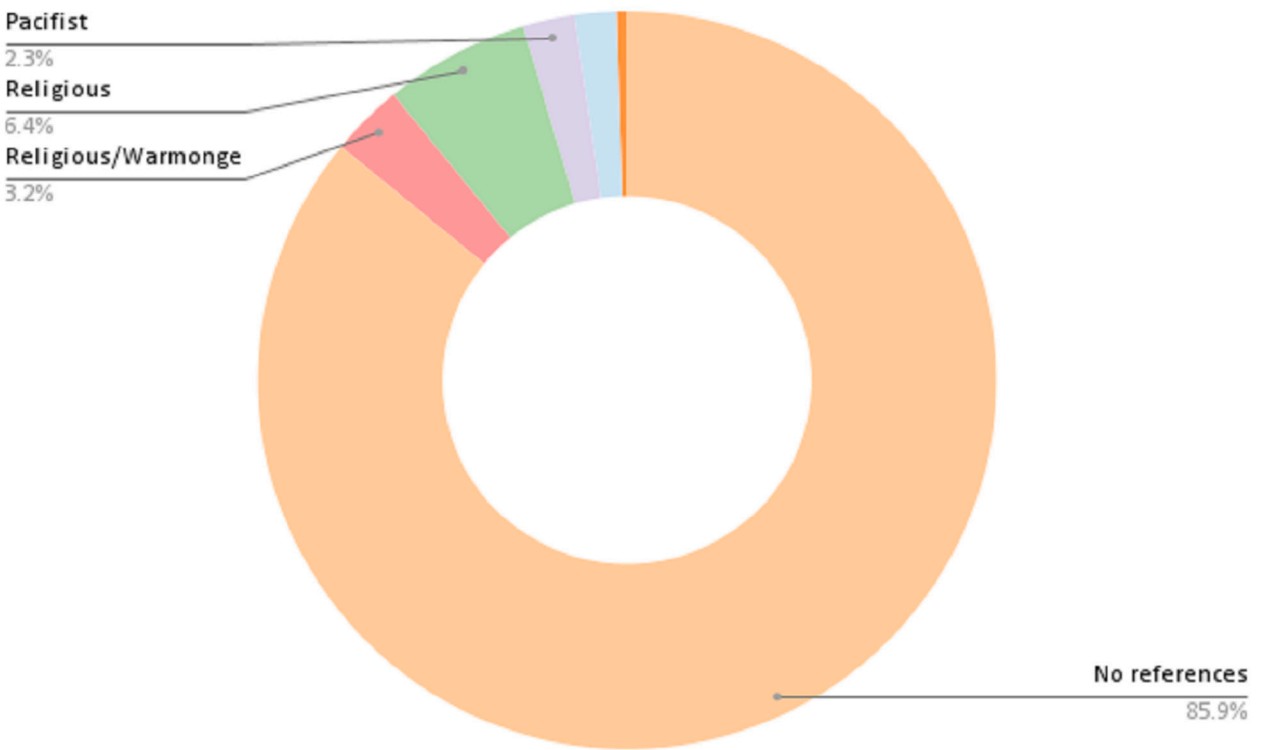

**Figure 7.** Doctrinal position on the Israel–Palestine conflict. Source: The authors' own creation.

## 4. Discussion and Conclusions

The methodology used in this study involved a series of steps, including data collection, assessment of videos' relevance, analysis of hashtag frequency, assessment of user authority, and exploration of the centrality of hashtags in a relational graph. These techniques provided detailed insight into the content dynamics on the social media platform studied, which has by far the most downloads in the world today (Statista 2023b). Although Tiktok is often associated with entertainment content, the data collected showed that the social network's users are not unaware of the news reality, as seen in the surge of comments about the conflict analyzed, and this reflects how social networks are playing an increasingly important role in news dissemination, both in positive terms, such as generating debate, and in negative terms (Marcos-García et al. 2021), such as contributing to polarization, disinformation, or manipulation. The patterns observed in TikTok users' posts illustrated the complexity of conversations about the Israel–Hamas conflict on the platform, conversations that address emotional, cultural, ethical, and legal issues and reflect the diversity of voices and perspectives on this social platform, as well as the presence of bias due to lacking the context and historical perspective necessary to make that conversation relevant (Newman et al. 2023).

The analysis reveals that there was a clear focus on the key actors of the conflict, in particular Palestine, Hamas, and Israel, and significant references to Islam more or less

directly and coming from Islamophobia (Civil et al. 2020). We were able to find a very polarized context, with comments clearly in favor of or against the main actors and with a tone that we have categorized as verbal aggression, that is, pithy comments with little reflective quality, which makes constructive debate difficult.

Of note was many users' insistence on differentiating between Hamas and Palestine, with clear support for the latter over the former. By contrast, among Israel's supporters, there was much less of this differentiation, although there was no explicit claim that Hamas and the Palestinian people as a whole are the same. Additionally, the mistrust toward the media as a means to understand the Israel–Palestine conflict in general and the current events stemming from the Hamas attack on Israel featured significantly. Independent of the mentions of Islam, references to religion in general and to "God" ["Dios"] were also significant, both in warmongering messages in support of Israel and in pacifist messages. In the vein of pacifism, we also found a deep concern for the respect of human rights, irrespective of the abovementioned actors. Finally, there were notable references to Islam, not only to religion but also to everything very generally associated with this religion, including a significant number of prejudices and clichés about Muslims (Bourekba 2018), such as the complete absence of women's rights in Muslim-majority societies and the use of derogatory terms such as "Moor" ["moro"] and offensive references such as "they do not eat ham" ["no comen jamón"] or a very derogatory and xenophobic one such as "Jovenlandia".

Of note was an example in one of the cases we found in our analysis. The case of the 40 babies supposedly beheaded by Hamas, which, despite the fact that it was not confirmed by the Israeli government itself, appeared many times in the TikTok comments analyzed, as mentioned above, and which is the best example of all that we analyzed and commented on regarding disintermediation, decontextualization, manipulation, dehumanization, and hate speech. This framework of implicit Islamophobia appeared in a large number of TikTok user comments strongly permeated by a series of clichés and prejudices about Islam. They presented a view of this religion that blurred together the practitioners of the religion and people of Arab ethnicity, regardless of their beliefs, as well as a multitude of cultural characteristics of Asia and North Africa; that is, it plays into a supposed troubled and contentious relationship between East and West (Said 1978). We observed how Muslims as a whole were repeatedly associated with terrorists and Islam was seen as incompatible with respect for human rights, as has been affirmed in research by various authors (Rahman 2022; Said 1978); this coincides with the monolithic representation of Islam as sexist, fanatical, homogeneous, and supportive of terrorism (Civil et al. 2023) as, for decades, it has been made out to be in Western media and now also on the social networks that very young audiences frequent (Gómez-Calderón et al. 2023).

Responding to the research questions initially posed, after this study that concurrently navigated a current event of particular relevance, we can conclude (RQ1) that young audiences were interested in the escalation of the conflict in the Middle East and that the polarized conversation on TikTok increased significantly. Similarly, the analysis of the extracted and filtered sample of comments (17,654) indicated that the variable "hate speech" (RQ2) had intensified in these young audiences' conversations, with about 21% of comments calling into question TikTok's own rules regarding user behaviors that incite hatred and that, within that percentage, there was a very large majority that generated and supported Islamophobic content. It is demonstrated, therefore, that there is an interest in this type of conversation about complex issues among young audiences, but that these conversations are not in line with said complexity, being quite superficial and sometimes offensive due to the polarization generated by misinformation. It is also evident that the platform's filters do not work properly, as hate speech is detected too frequently.

**Author Contributions:** Conceptualization, J.-L.G.-E.; Methodology, L.M.-P., C.M.L.-R. and F.S.-Q.; Validation, L.M.-P., C.M.L.-R. and F.S.-Q.; Data curation, L.M.-P., C.M.L.-R. and F.S.-Q.; Writing, J.-L.G.-E.; Review and editing, J.-L.G.-E. and C.M.L.-R.; Supervision, J.-L.G.-E. and C.M.L.-R. All authors have read and agreed to the published version of the manuscript.

**Funding:** National project (Ministry of Science and Innovation–Government of Spain: "Young Spaniards' use of social networks for news: Incidental news consumption, technological conditioning factors, and credibility of journalistic content" ["El uso informativo de las redes sociales por parte de los jóvenes españoles: consumo incidental de noticias, condicionantes tecnológicos y credibilidad de los contenidos periodísticos"]. Ref. PID2019-106932RB-100).

**Institutional Review Board Statement:** Not applicable.

**Informed Consent Statement:** Not applicable.

**Data Availability Statement:** The data presented in this study are openly available in Gonzalez Esteban, Jose Luis; Lopez-Rico, Carmen M; Morales Pino, Loraine; Sabater Quinto, Federico (2024). TikTok Comments about Palestine and Israel war 23 Data Base.xlsx. figshare. Dataset. https://figshare.com/articles/dataset/TikTok_Comments_about_Palestine_and_Israel_war_23_Data_Base_xlsx/24981285/1 (accessed on 12 October 2023).

**Conflicts of Interest:** The authors declare no conflicts of interest.

## Note

1    https://github.com/bellingcat/tiktok-hashtag-analysis (accessed on 12 October 2023).

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
