# Peer review of "Intensification of Hate Speech, Based on the Conversation Generated on TikTok during the Escalation of the War in the Middle East in 2023"

_socsci, doi:10.3390/socsci13010049_

Round 1

Reviewer 1 Report

Comments and Suggestions for Authors

The article addresses a topic of great interest - hate speech - and offers an original analysis of a still little-studied platform like TikTok. The premises and the literary review are adequate and take into consideration different theoretical backgrounds useful for the interpretation of the phenomenon. The methodology is adequate and well described, also presenting the limits of the choices made.

The conclusion could advance some further interpretations, while remaining on a descriptive level.

Author Response

Thank you very much for your comment, we take note to advance the conclusions of those additional interpretations.

Reviewer 2 Report

Comments and Suggestions for Authors

This is an interesting and timely study. There are urgent revisions to be made, however, and the paper is not ready for publication yet. I hope that the comments below will be useful with the revisions suggested.

1. The paper starts with the concept of 'disintermediation' which is rather confusing, as it is a business and finance concept, unrelated to politics or hate speech. Do you perhaps mean disinformation, or something else? Please revise this.

2. The first paragraph, and indeed other aspects of the text, lack synthesis. Specifically, the works of Habermas on the theory of communicative action, scholarly work on techno-populism, otherness, infotainment, virtual reality, public sphere fragmentation, aporophobia, and others, are bundled together without a coherent thread. Section 1.2 is equally problematic, and these parts need to be rewritten with a clearer structure and synthesis. Moving from the general to the specific would be a good suggestion, for example.

3. The research questions are rather confusing, as the paper seems to focus on hateful communication in Spanish on TikTok related to the Israel-Hamas conflict. The RQs are more generic and not specific enough. 

4. How are young audiences determined on TikTok, and how do the researchers know which posts and comments were made by young audiences? This refers to RQ2, which again needs to be reconsidered. 

5. Points of bias: Why is it necessary to specify that Emcke is a German author? Why is it necessary to describe TikTok as "the Chinese platform"? Why refer to research on "alleged" Spanish antisemitism, where there is published research on this? The paper seems to imply that there is no Spanish antisemitism, for example, which is a bold claim to make. Research on Spanish antisemitism would suffice here, for example. 

6. Methodology:  "To perform a qualitative analysis of the comments on the discussions of these videos, 268 we made sure that they had at least 60 likes, since we knew that such statements had 269 considerable support from viewers." Why 60 likes? This seems rather arbitrary, unless you can cite similar studies and demonstrate that this is the typical threshold or benchmark used. 

7. Method: The study mentions the method of data collection and selection with scraping techniques, but not the method of analysis. How was the data analysed after it was collected? 

8. The discussion of results seems to suggest that the results are generalisable ("public opinion suggests"...) which is not the case. This is perhaps a question of framing, but any results are valid for the specific sample, and not for public opinion in general. 

9. There is no mention that the study has received ethical approval from an institution or funder. Please include this in the revised version of the manuscript.

10. The paper would benefit from proofreading as there are grammar errors, over-long sentences and complex syntactical constructions. 

11. The paper would benefit from a wider bibliography on hate speech and cyberhate in particular, as well as similar case studies from other countries or groups. 

Comments on the Quality of English Language

Please see point 10 above.

Author Response

Thank you very much for taking the time to review this manuscript. Please find the detailed responses below and the corresponding revisions/corrections highlighted/in track changes in the re-submitted file,

1) The paper starts with the concept of 'disintermediation' which is rather confusing, as it is a business and finance concept, unrelated to politics or hate speech. Do you perhaps mean disinformation, or something else? Please revise this.

Yes, we are referring to the transformations of intermediation, in the political sphere and, fundamentally, in the journalistic sphere. What previously could only be achieved through intermediaries, today can be done autonomously thanks to new technologies and this is a core issue in the research at hand. In any case, the concept has been revised and better explained.

2) The first paragraph, and indeed other aspects of the text, lack synthesis. Specifically, the works of Habermas on the theory of communicative action, scholarly work on techno-populism, otherness, infotainment, virtual reality, public sphere fragmentation, aporophobia, and others, are bundled together without a coherent thread. Section 1.2 is equally problematic, and these parts need to be rewritten with a clearer structure and synthesis. Moving from the general to the specific would be a good suggestion, for example.

The suggested contributions have been taken into account and some concepts have been reordered and better explained. The ideas have been better structured, ranking from the most general to the most specific issues. The writing has been revised and the bibliography has been expanded.

3) The research questions are rather confusing, as the paper seems to focus on hateful communication in Spanish on TikTok related to the Israel-Hamas conflict. The RQs are more generic and not specific enough. 

The recommendations have been taken into account and the research questions have been reformulated to clearly convey the fundamental issues that are addressed and intended to be resolved in this research.

4) How are young audiences determined on TikTok, and how do the researchers know which posts and comments were made by young audiences? This refers to RQ2, which again needs to be reconsidered.

We determined that the comments are from young people since this social network is mostly used by an audience under 17 years of age. In 2023, 48% of its users are between 12 and 24 years old and 57% under 34. We incorporate bibliography that indicates it.

5) Points of bias: Why is it necessary to specify that Emcke is a German author? Why is it necessary to describe TikTok as "the Chinese platform"? Why refer to research on "alleged" Spanish antisemitism, where there is published research on this? The paper seems to imply that there is no Spanish antisemitism, for example, which is a bold claim to make. Research on Spanish antisemitism would suffice here, for example.

In this sense we simply refer to their origin with no other intention than to provide one more piece of information to the reader and as a resource in the writing to avoid repeating the name of the social network or the authors. However, following your recommendation and  in order not to offend anyone, they have been removed. Regarding anti-Semitism, the study does not intend to prove that it exists in Spain or not, we want to see the hate speech that exists among young people, Tik Tok is the network with younger users in general, towards Israel or Palestine since it is a fact of strict topicality at this time.  Besides, a study by Córdoba-Hernández, 2011 is cited, in which the positioning of Spanish society and its different governments since the Franco era is explained as a context, since support for Israel or Palestine has been varying. Our results also show that there are insults/hate and support for both nations. In any case, we understand the recommendation and proceed to improve the texts.

6) Methodology:  "To perform a qualitative analysis of the comments on the discussions of these videos, 268 we made sure that they had at least 60 likes, since we knew that such statements had 269 considerable support from viewers." Why 60 likes? This seems rather arbitrary, unless you can cite similar studies and demonstrate that this is the typical threshold or benchmark used. 

In this sense, discarding the comments that did not have any likes because they have not had interaction from anyone, we consider that a sample of more than 200 random video comments is enough to show the feeling on the topic. In a strictly quantitative sense, it is double the median, which is 33 likes.

7) Method: The study mentions the method of data collection and selection with scraping techniques, but not the method of analysis. How was the data analysed after it was collected?

We have added information about Speech analysis in the methodology section to clarify the reviewers' doubts. Figure number 3 has been updated and improved

8) The discussion of results seems to suggest that the results are generalisable ("public opinion suggests"...) which is not the case. This is perhaps a question of framing, but any results are valid for the specific sample, and not for public opinion in general. 

We have corrected the text in this sense. However, during the coding process, the 'public opinion' code was defined to refer to the users' opinions on the topics addressed in the video and their relationship with other codes, as shown in Figure 3. That is, after analyzing the selected comments we can establish that the users' opinion is directly related to sarcasm, misinformation, verbal aggression, religious or faith elements, and messages of support. Furthermore, Human Rights, comparisons, support for Palestine and disagreement were part of the opinion expressed in the social network publications. We also changed public opinion code name to opinion in Figure 3 and the text to avoid confusion.

9) There is no mention that the study has received ethical approval from an institution or funder. Please include this in the revised version of the manuscript.

This research is similar to other studies published in the journal, which have not had such approval either since they have not worked with sensitive or personal data.

10) The paper would benefit from proofreading as there are grammar errors, over-long sentences and complex syntactical constructions.

This suggestion has been taken into account and some phrases that could be confusing from a grammatical point of view have been corrected. In addition, we have suggested a final review to our professional translators from our research group to avoid linguistic errors or typos.

11) The paper would benefit from a wider bibliography on hate speech and cyberhate in particular, as well as similar case studies from other countries or groups. 

Following your recommendation we have expanded the bibliography.

Thank you very much for your attention